# Generalized estimating equation modeling on correlated microbiome sequencing data with longitudinal measures

**Bo Chen**[1], **Wei Xu**[1,2]*

**1** Princess Margaret Hospital, Toronto, Ontario, Canada, **2** Dalla Lana School of Public Health, University of Toronto, Toronto, Ontario, Canada

* wei.xu@uhnresearch.ca

**Data Availability Statement:** All relevant data are within the manuscript and its Supporting Information files.

**Funding:** WX was funded by Natural Sciences and Engineering Research Council of Canada (NSERC

## Abstract

Existing models for assessing microbiome sequencing such as operational taxonomic units (OTUs) can only test predictors' effects on OTUs. There is limited work on how to estimate the correlations between multiple OTUs and incorporate such relationship into models to evaluate longitudinal OTU measures. We propose a novel approach to estimate OTU correlations based on their taxonomic structure, and apply such correlation structure in Generalized Estimating Equations (GEE) models to estimate both predictors' effects and OTU correlations. We develop a two-part Microbiome Taxonomic Longitudinal Correlation (MTLC) model for multivariate zero-inflated OTU outcomes based on the GEE framework. In addition, longitudinal and other types of repeated OTU measures are integrated in the MTLC model. Extensive simulations have been conducted to evaluate the performance of the MTLC method. Compared with the existing methods, the MTLC method shows robust and consistent estimation, and improved statistical power for testing predictors' effects. Lastly we demonstrate our proposed method by implementing it into a real human microbiome study to evaluate the obesity on twins.

## Author summary

Human microbiome sequencing data analysis has been a fast growing area of genomic research in recent years. Although there have been several works for detecting predictors on a single operational taxonomic unit (OTU) or multiple OTUs simultaneously, there is limited work on how to estimate the correlations between multiple OTUs and incorporate such relationship into models to evaluate longitudinal OTU measures. Here we propose a novel approach to estimate OTU correlations based on their taxonomic structure after integrating longitudinal and other types of repeated OTU measures, and apply such correlation structure in Generalized Estimating Equations (GEE) models to estimate both predictors' effects and OTU correlations. The method is theoretically sound and practically easy to implement, and we provide corroborating evidence from simulation and a real human microbiome study.

Grant RGPIN-2017-06672), Princess Margaret
Cancer Foundation Award. BC is a post-doctoral
fellowship trainee and supported by Princess
Margaret Cancer Foundation for AI and
Microbiome Program. The funders had no role in
study design, data collection and analysis, decision
to publish, or preparation of the manuscript.

**Competing interests:** The authors have declared
that no competing interests exist.

This is a *PLOS Computational Biology* Methods paper.

## Introduction

Human microbiome sequencing data analysis has been a fast-growing area of genomic
research in recent years. Several studies showed that the microbial composition is associated
with environmental and host factors [1–3]. The microbiome data are usually characterized by
16S ribosomal ribonucleic acid (rRNA) gene sequencing or shotgun metagenomics sequencing
[4, 5]. Both sequencing technologies provide reads of bacteria counts clustered into operational
taxonomic units (OTUs), where each OTU is typically mapped to a taxon at level species,
genus, family, order, class, phylum, kingdom or domain in a taxonomic structure.

For each sample, OTU counts can be converted to relative abundances (RAs). No matter
the OTU data is in format of counts or RAs, there are a few analytical challenges which prevent
the application of standard regression methods on association study between microbial com-
position and the environmental or genetic factors. First, the OTU data usually contains exces-
sive zeros, which prevents modelling the OTU data by using standard types of distributions.
Next, for each individual, there may exist repeated measures of OTUs, such as microbiome
samples collected from different locations of human body, or multiple observations at different
time points in longitudinal setting. Furthermore, the sequencing method usually detects hun-
dreds or thousands of OTUs, which are potentially correlated with each other [6]. Identifying
correlations between taxa is a common goal in genomic survey [7]. An accurate estimated cor-
relation can be used to determine drivers in environmental ecology or contribution to habitat
niches or disease; it is also a powerful tool to help researchers with hypothesis generation, such
as determining which interactions might be biologically relevant in their system, and should
be given further study [8]. So instead of considering each OTU as independent, it is desirable
to incorporate the taxonomic information into the analysis, which reflects the correlation
structure between the OTUs.

Several solutions have been proposed to answer each of these challenges. Zero-inflated
microbiome data can be fitted by either zero-inflated models or two-part models [9, 10].
Repeated measures can be characterized by random effects in mixed effects models [11–15].
Modelling multiple OTUs together remains a challenging problem, although several attempts
have been made. La Rosa et al. [16] and Chen et al. [17] proposed an approach which assumes
that multiple OTUs follow Dirichlet multinomial (DM) distribution. However, the DM
assumption imposes a negative correlation among OTUs where the true correlation can be
both positive and negative. In addition, it has a fixed covariance structure which cannot flexi-
bly handle various dispersion patterns. Tang et al. [18] proposed zero-inflated generalized
Dirichlet multinomial distribution which allows for a more general covariance structure and
excessive zeros in OTU counts. To further eliminate the negative correlation assumption, they
also proposed distribution-free non-parametric tests [19, 20], which are robust to any correla-
tion structures within a cluster of taxa. However, parameter estimates of covariate effects and
correlation coefficients were not available due to the non-parametric essence. Alternatively,
Shi et al. [21] proposed a model for Paired-Multinomial Data which works for a pair of
repeated measures or a pair of correlated OTUs. Zhang at al. [22] considered estimating pair-
wise correlations between OTUs. Xu et al. [23] used latent variables to account for the correla-
tion of multiple OTUs. Zhan et al. and Koh at al. [24, 25] adopted correlated sequence kernel
association test assuming a random effect for each OTU, and Grantham et al. [26] used Bayes-
ian factor analysis to cluster correlated OTUs into different factors. However, none of these

approaches can model the taxonomic relationship between OTUs and provide estimations for complex correlation structure.

In order to estimate and test the association between the predictors and OTUs as well as simultaneously estimating the correlation parameters between OTUs, we propose a generalized estimating equation (GEE) [27] approach which can handle multiple correlated OTUs with repeated measures. Applying GEE model to either microbiome data [28, 29] or repeated measures such as longitudinal zero-inflated data [30–32] is not new. The novel part of our method is to develop and construct correlation structures which can truly represent the taxonomic correlations and time dependency of longitudinal OTU measures. First, we develop a correlation structure of multiple OTUs solely depending on their taxonomic structure, so that the correlation structure can provide meaningful estimates of OTU correlations. Not like the multinomial models which assume negative correlations, the correlation of OTUs in the proposed model can be both positive and negative. In addition, we incorporate the taxonomic structure with correlations due to repeated measures, and all correlations of repeated measures can be explicitly estimated.

We organize this paper as following. In Methodology section, the detailed methodology framework is introduced including the zero-inflated GEE models, the construction of correlation structure on multiple OTUs with repeated measures, parameter estimation and hypothesis testing under the Microbiome Taxonomic Longitudinal Correlation (MTLC) model. Extensive simulation studies for comparing the performance of the proposed approach to other models are presented in Simulation section. In Application section, the proposed model is applied into a real microbiome sequencing study. The conclusion and further improvements of our method are discussed in Discussion section.

## Methodology

### Taxonomic structure of OTUs

**Numerical representation of taxonomic structure.** For known taxonomic structure of $N$ OTUs, we consider its numeric representation, i.e., representing the structure by a list of numerical vectors. Throughout this paper, we call taxonomic levels from species to domain from lowest to highest. First, we find the taxonomic level at which all observed $N$ OTUs belong to the same taxon but not at one level lower, and define such level as level 1. For example, if all OTUs belong to the same class but not the same order, then the level class would be level 1. Similarly, we can identify the taxonomic level at which each OTU represents a different taxon but not at one level higher, and define such level as level $I$. For example, if each OTU belongs to a different genus but not a different family, then the level genus would be level $I$. Fig 1 illustrates an example with $I = 4$ (class, order, family, and genus), where class is level 1 and genus is level 4.

For $i = 1, \ldots I$, let $M_i$ be the number of taxa at taxonomic level $i$. By definition, $M_1 = 1$ and $M_i = N$. For $m_i = 1, \ldots, M_i$, $t_{m_i i}$ denotes each taxon at level $i$, and $n_{m_i i}$ is the number of OTUs belonging to taxon $t_{m_i i}$. $n_{m_i i}$ are then computed by the following algorithm:

1. When $i = I$, $n_{m_i i} = 1$.

2. For $i = I - 1, \ldots, 1$,

$$n_{m_i i} = \sum_{t_{m_{i+1} i+1} \in t_{m_i i}} n_{m_{i+1} i+1}.$$

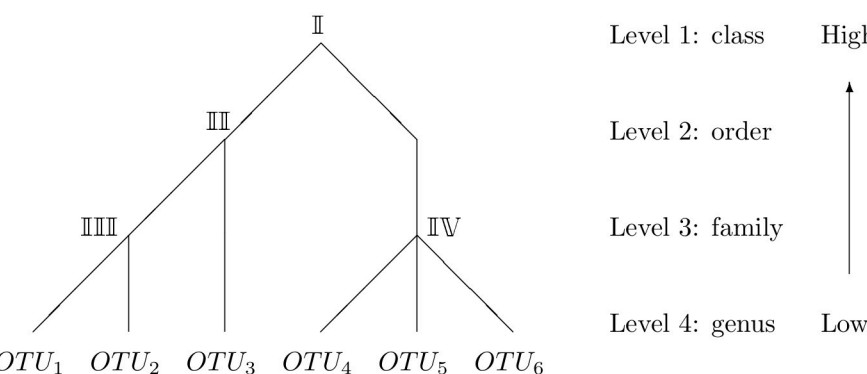

**Fig 1. Example illustrating the taxonomic structure of 6 hypothetical OTUs.**

It is easy to check that for $i = 1, \ldots, I$,

$$\sum_{m_i=1}^{M_i} n_{m_i i} = N.$$

Let $\boldsymbol{n_i} = (n_{1i}, \ldots, n_{M_i i})$. Then the taxonomic structure can be numerically represented by $(\boldsymbol{n_1}, \ldots, \boldsymbol{n_I})$.

In the illustrative taxonomic structure example from Fig 1, we observe 6 correlated OTUs with $I = 4$. Then $M_1 = 1, M_2 = 2, M_3 = 3, M_4 = 6$, and the numerical representation of Fig 1 is $\boldsymbol{n_1} = 6, \boldsymbol{n_2} = (3, 3), \boldsymbol{n_3} = (2, 1, 3), \boldsymbol{n_4} = (1, 1, 1, 1, 1, 1)$.

**Correlation matrix of taxonomic structure.** Following the taxonomic structure, it is natural to assume that OTUs belonging to same taxa at higher levels may have some correlation. Because all OTUs belong to the same taxa at the highest taxonomic level (e.g., Bacteria domain), they are all correlated in principle. For $N$ OTUs, there are up to $\binom{N}{2}$ pairwise correlations. When $N$ is large, it would be infeasible to model $\binom{N}{2}$ correlation parameters, and our intuition is to reduce the number of parameters by making some reasonable assumptions such that many of the correlations are equal, according to the known taxonomic structure. The basic assumption we made is that for a cluster of OTUs, if each OTU represents a different taxon at level $i + 1$ but they all belong to the same taxon at level $i$, then all pairwise correlations of OTUs within this cluster should be equal. Under this assumption, there is only one correlation parameter in the simple case when $I = 2$. When $I > 2$, there are more than two levels in the OTU taxonomic structure, in which case the pairwise correlation coefficients for different pairs of OTUs may be equal or unequal, depending on the taxa which the OTUs belong to at each level. For a pair of OTUs, if they belong to different taxa at level $i + 1$ but the same taxa at level $i$, we call the taxon at level $i$ as its first common taxon. For any two pairs of OTUs. A natural extension of our basic assumption is that two pairs of OTUs are assumed to have same correlation if and only if the first common taxa of both pairs are identical. Formally, let $\mathcal{P}^*$ and $\mathcal{P}^\dagger$ be two pairs of OTUs, which have correlation $\rho^*$ and $\rho^\dagger$. $t_{m_i^*, i^*}$ is the first common taxon of $\mathcal{P}^*$, and $t_{m_i^\dagger, i^\dagger}$ is the first common taxon of $\mathcal{P}^\dagger$. Then we assume

$$\rho^* = \rho^\dagger \Longleftrightarrow t_{m_i^*, i^*} = t_{m_i^\dagger, i^\dagger}$$

For all $N$ OTUs, we define a taxonomic structure matrix to indicate which correlations are equal and which are not. The taxonomic structure matrix is an $N \times N$ symmetric matrix, where all diagonal entries are denoted by $\mathbb{D}$, and off-diagonal entries are indexed by uppercase Roman numbers, i.e., $\mathbb{I}$, $\mathbb{II}$, $\mathbb{III}$ (see Fig 1). Each different index value represents a different correlation, and equal index value indicates the corresponding correlations are estimated by the same coefficient. We use Roman numbers to avoid any confusion with other Arabic numerals used elsewhere throughout our work, because these indices are categorical numbers which do not indicate any quantity. The values of off-diagonal entries are determined by the following steps:

1. For $i = 1, \ldots, I - 1$, Let $\mathbf{\Gamma}_i$ be an $N \times N$ block diagonal matrix,

$$\mathbf{\Gamma}_i = \begin{pmatrix} \mathbf{B}_{1i} & & \\ & \ddots & \\ & & \mathbf{B}_{M_i i} \end{pmatrix}.$$

For $m_i = 1, \ldots, M_i$, each block $\mathbf{B}_{1i}$ is an $n_{m_i i} \times n_{m_i i}$ matrix, whose diagonal entries are $\mathbb{D}$ and off-diagonal entries are $\sum_{h=0}^{i-1} M_h + m_i$. $M_0$ has default value 0.

2. When $i = 1$, Let $\mathbf{\Gamma}^{(1)} = \mathbf{\Gamma}_1$ be the interim correlation matrix.

3. When $i = 2, \ldots, I - 1$, replace the block diagonal entries of $\mathbf{\Gamma}^{(i-1)}$ by $\mathbf{B}_{m_i i}$ and keep all other entries the same. The interim correlation matrix after the replacement at level $i$ is defined as $\mathbf{\Gamma}^{(i)}$.

4. Sort all off-diagonal entries in $\mathbf{\Gamma}^{(I-1)}$ from largest to smallest, where the smallest value corresponds to smallest order (order 1). Replace all off-diagonal entries by their corresponding orders in uppercase Roman numbers and define the new matrix as $\mathbf{\Gamma}$. $\mathbf{\Gamma}$ is the taxonomic structure matrix which is numerically represented by $(\mathbf{n}_1, \ldots, \mathbf{n}_I)$.

In the above example of 6 hypothetical OTUs in Fig 1,

$$\mathbf{\Gamma}_1 = \begin{pmatrix} \mathbb{D} & \mathbb{1} & \mathbb{1} & \mathbb{1} & \mathbb{1} & \mathbb{1} \\ \mathbb{1} & \mathbb{D} & \mathbb{1} & \mathbb{1} & \mathbb{1} & \mathbb{1} \\ \mathbb{1} & \mathbb{1} & \mathbb{D} & \mathbb{1} & \mathbb{1} & \mathbb{1} \\ \mathbb{1} & \mathbb{1} & \mathbb{1} & \mathbb{D} & \mathbb{1} & \mathbb{1} \\ \mathbb{1} & \mathbb{1} & \mathbb{1} & \mathbb{1} & \mathbb{D} & \mathbb{1} \\ \mathbb{1} & \mathbb{1} & \mathbb{1} & \mathbb{1} & \mathbb{1} & \mathbb{D} \end{pmatrix}, \mathbf{\Gamma}_2 = \begin{pmatrix} \mathbb{D} & \mathbb{2} & \mathbb{2} & & & \\ \mathbb{2} & \mathbb{D} & \mathbb{2} & & & \\ \mathbb{2} & \mathbb{2} & \mathbb{D} & & & \\ & & & \mathbb{D} & \mathbb{3} & \mathbb{3} \\ & & & \mathbb{3} & \mathbb{D} & \mathbb{3} \\ & & & \mathbb{3} & \mathbb{3} & \mathbb{D} \end{pmatrix},$$

$$\mathbf{\Gamma}_3 = \begin{pmatrix} \mathbb{D} & \mathbb{4} & & & & \\ \mathbb{4} & \mathbb{D} & & & & \\ & & \mathbb{D} & & & \\ & & & \mathbb{D} & \mathbb{6} & \mathbb{6} \\ & & & \mathbb{6} & \mathbb{D} & \mathbb{6} \\ & & & \mathbb{6} & \mathbb{6} & \mathbb{D} \end{pmatrix}.$$

Applying step 2 and 3 to achieve

$$\mathbf{\Gamma^{(3)}} = \begin{pmatrix} \mathbb{D} & 4 & 2 & 1 & 1 & 1 \\ 4 & \mathbb{D} & 2 & 1 & 1 & 1 \\ 2 & 2 & \mathbb{D} & 1 & 1 & 1 \\ 1 & 1 & 1 & \mathbb{D} & 6 & 6 \\ 1 & 1 & 1 & 6 & \mathbb{D} & 6 \\ 1 & 1 & 1 & 6 & 6 & \mathbb{D} \end{pmatrix}$$

Applying step 4 and the final taxonomic structure matrix $\mathbf{\Gamma}$ is

|        | $OTU_1$ | $OTU_2$ | $OTU_3$ | $OTU_4$ | $OTU_5$ | $OTU_6$ |
|--------|---------|---------|---------|---------|---------|---------|
| $OTU_1$ | $\mathbb{D}$ | $\mathbb{III}$ | $\mathbb{II}$ | $\mathbb{I}$ | $\mathbb{I}$ | $\mathbb{I}$ |
| $OTU_2$ | $\mathbb{III}$ | $\mathbb{D}$ | $\mathbb{II}$ | $\mathbb{I}$ | $\mathbb{I}$ | $\mathbb{I}$ |
| $OTU_3$ | $\mathbb{II}$ | $\mathbb{II}$ | $\mathbb{D}$ | $\mathbb{I}$ | $\mathbb{I}$ | $\mathbb{I}$ |
| $OTU_4$ | $\mathbb{I}$ | $\mathbb{I}$ | $\mathbb{I}$ | $\mathbb{D}$ | $\mathbb{IV}$ | $\mathbb{IV}$ |
| $OTU_5$ | $\mathbb{I}$ | $\mathbb{I}$ | $\mathbb{I}$ | $\mathbb{IV}$ | $\mathbb{D}$ | $\mathbb{IV}$ |
| $OTU_6$ | $\mathbb{I}$ | $\mathbb{I}$ | $\mathbb{I}$ | $\mathbb{IV}$ | $\mathbb{IV}$ | $\mathbb{D}$ |

In taxonomic structure matrix $\mathbf{\Gamma}$, the index values are illustrated in Fig 1: index $\mathbb{I}$ indicates correlation of OTUs belonging to the same class but different orders; index $\mathbb{II}$ indicates correlation of OTUs belonging to the same order but different families; index $\mathbb{III}$ and $\mathbb{IV}$ indicate correlations of OTUs belonging to the same family.

## Modelling correlations from repeated measures

**Correlations of longitudinal data.** Repeated measures of single OTU from the same individual may be another source of correlation, e.g., OTU observation at multiple time points within the same person. Fig 2 shows repeated measures of multiple OTUs at $l$ time points.

There are several different ways to characterize the correlations between each pair of time points, such as exchangeable, Toeplitz and unstructured. Exchangeable structure assumes all correlations are equal to each other. Toeplitz structure assumes time points with equal temporal distance have equal correlation. Unstructured model assumes each pair has different correlations and it is the most complicated structure in terms of correlation parameter estimation. Besides that, other correlation structures such as autoregressive, moving averages are also used for longitudinal data analysis [33, 34]. In this paper, we assume the correlation structure within the same individual is pre-specified. The correlation structure matrix within same individual following a given correlation structure is denoted by $\Omega_T$. The diagonal entries are denoted by

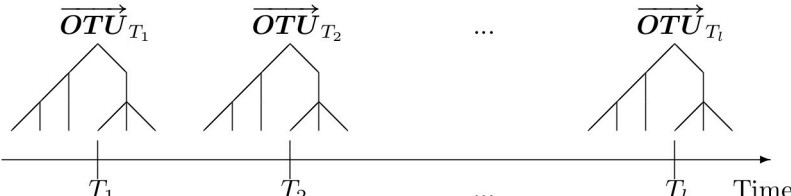

**Fig 2. Longitudinal OTU observations at $l$ time points.**

$\mathbb{D}$ again, and off-diagonal entries are indexed by lowercase Roman numbers, i.e., i, ii, iii, etc.. For example, if the longitudinal OTU observations consist of 3 time points, then $\Omega_T$ assuming exchangeable structure is

|       | $T_1$        | $T_2$        | $T_3$        |
|-------|--------------|--------------|--------------|
| $T_1$ | $\mathbb{D}$ | i            | i            |
| $T_2$ | i            | $\mathbb{D}$ | i            |
| $T_3$ | i            | i            | $\mathbb{D}$ |

Alternatively, $\Omega_T$ assuming Toeplitz structure is

|       | $T_1$        | $T_2$        | $T_3$        |
|-------|--------------|--------------|--------------|
| $T_1$ | $\mathbb{D}$ | i            | ii           |
| $T_2$ | i            | $\mathbb{D}$ | i            |
| $T_3$ | ii           | i            | $\mathbb{D}$ |

**Sample correlation.** In addition to time correlation, there may exist other types of sample correlations, such as two or more individuals from the same pedigree, or simply any repeated measures from the same individual. Without loss of generality we assume there are two repeated samples $S_1$ and $S_2$. Then sampling correlation is represented by correlation structure matrix $\Omega_S$:

|       | $S_1$        | $S_2$        |
|-------|--------------|--------------|
| $S_1$ | $\mathbb{D}$ | i            |
| $S_2$ | i            | $\mathbb{D}$ |

**Combining longitudinal and sample correlation.** Let $\Omega$ be the correlation structure combining both longitudinal and sample correlation. $\Omega = \Omega_T$ or $\Omega_S$ when only time points correlation or sample correlation exists. When both correlations exist, we consider all combinations of time points and repeated samples in one big correlation structure $\Omega$. For example, if there are two repeated samples at each of the 3 time points, then for each OTUs there are 6 observations for each individual in total, and $\Omega$ becomes

|              | $(T_1, S_1)$ | $(T_2, S_1)$ | $(T_3, S_1)$ | $(T_1, S_2)$ | $(T_2, S_2)$ | $(T_3, S_2)$ |
|--------------|--------------|--------------|--------------|--------------|--------------|--------------|
| $(T_1, S_1)$ | $\mathbb{D}$ | i            | i            | ii           | iii          | iii          |
| $(T_2, S_1)$ | i            | $\mathbb{D}$ | i            | iii          | ii           | iii          |
| $(T_3, S_1)$ | i            | i            | $\mathbb{D}$ | iii          | iii          | ii           |
| $(T_1, S_2)$ | ii           | iii          | iii          | $\mathbb{D}$ | i            | i            |
| $(T_2, S_2)$ | iii          | ii           | iii          | i            | $\mathbb{D}$ | i            |
| $(T_3, S_2)$ | iii          | iii          | ii           | i            | i            | $\mathbb{D}$ |

## Incorporating taxonomic structure with repeated measures

Suppose $\Omega$ has dimension $L$. For $a = 1, \ldots, N$ and $b = 1, \ldots, N$, $\Omega(\Gamma_{ab})$ is an $L \times L$ correlation matrix as a function of $\Gamma_{ab}$, such that

$$\boldsymbol{\Omega}(\Gamma_{ab}) = \begin{pmatrix} \rho_{(\Gamma_{ab},\Omega_{11})} & \cdots & \rho_{(\Gamma_{ab},\Omega_{1L})} \\ \vdots & \ddots & \vdots \\ \rho_{(\Gamma_{ab},\Omega_{L1})} & \cdots & \rho_{(\Gamma_{ab},\Omega_{LL})} \end{pmatrix}.$$

$\Gamma_{..}$ and $\Omega_{..}$ are entries of $\boldsymbol{\Gamma}$ and $\boldsymbol{\Omega}$ from corresponding rows and columns. We denote $\Omega(\Gamma_{ab})$ as $\Omega^{ab}$ for notation simplicity.

To integrate repeated measures correlation structure $\Omega$ with taxonomic structure $\boldsymbol{\Gamma}$, we introduce the integrative correlation matrix

$$\boldsymbol{R} = \begin{pmatrix} \boldsymbol{\Omega}^{11} & \cdots & \boldsymbol{\Omega}^{1N} \\ \vdots & \ddots & \vdots \\ \boldsymbol{\Omega}^{N1} & \cdots & \boldsymbol{\Omega}^{NN} \end{pmatrix}$$

where $\Omega^{ab}$ is defined above. $\boldsymbol{R}$ is a $J \times J$ matrix where $J = N \times L$, and each of its entry has the form $\rho_{(\Gamma_{..},\Omega_{..})}$. The first subscript, $\Gamma_{..}$, is either $\mathbb{D}$ or an uppercase Roman number indexing taxonomic structure correlation; the second subscript, $\Omega_{..}$, is either $\mathbb{D}$ or a lowercase Roman number indexing correlation from repeated measures of single OTU. In the above example, $\Gamma_{11} = \Omega_{11} = \mathbb{D}$, $\Gamma_{21} = \mathbb{III}$ and $\Omega_{21} = \mathbb{i}$. The diagonal entries of $\boldsymbol{R}$, $\rho_{(\mathbb{D},\mathbb{D})}$ always equal to 1, and the off-diagonal entries are estimated in the next section.

## Microbiome Taxonomic Longitudinal Correlation (MTLC) model

After specifying the correlation matrix within one cluster of OTUs with repeated measures, in this section, we introduce how to model the association between multiple OTUs and their predictors of interest. We propose a Microbiome Taxonomic Longitudinal Correlation (MTLC) model to estimate predictor effects, correlation coefficients between OTUs, longitudinal measures and other repeated measures. We also perform a hypothesis testing of the predictor effects based on MTLC model. The estimates and tests are achieved by Generalized Estimating Equations (GEE) framework.

**Generalized estimating equation framework.** Let $\boldsymbol{y_k}$'s be independent clusters for $k = 1, \ldots K$, and each cluster $\boldsymbol{y_k} = (y_{k1}, \ldots y_{kJ_k})$ has length $J_k$. For $j = 1, \ldots J_k$, let $\boldsymbol{x_{kj}}$ denote the vector of covariates with length $p$, and $\boldsymbol{\mu_k} = (\mu_{k1}, \ldots, \mu_{kJ_k})$ is the mean of $\boldsymbol{y_k}$. Then for each observation $y_{kj}$,

$$g(\mu_{kj}) = \boldsymbol{x_{kj}}'\boldsymbol{\beta} \tag{1}$$

where $g$ is a known link function and $\boldsymbol{\beta}$ are the regression parameters of the $p$ covariates $\boldsymbol{x_{kj}}$. The conditional variance of $y_{kj}$ is defined as $\text{Var}(y_{kj}|\boldsymbol{x_{kj}}) = v(\mu_{kj})\phi$, where $v$ is the variance function depending on the distribution of $y_{kj}$, and $\phi$ is the dispersion parameter being $\sigma^2$ for normally distributed $y_{kj}$ and 1 for other distributions belonging to exponential family. For estimating $\boldsymbol{\beta}$, the following generalized estimating equation is solved:

$$U(\boldsymbol{\beta}) = \Sigma_{k=1}^K \boldsymbol{D_k}'\boldsymbol{V_k}^{-1}(\boldsymbol{y_k} - \boldsymbol{\mu_k}) = 0 \tag{2}$$

where $D_k = \frac{d\mu_k}{d\beta}$ and $V_k = A_k^{1/2} R_k(\rho) A_k^{1/2}$. Here $A_k = diag(\mu_{k1}\phi, \ldots \mu_{kJ_k}\phi)$, and $R_k(\rho)$ is the working correlation matrix following the correlation structure $R$ constructed in section "Incorporating taxonomic structure with repeated measures", where $\rho$ is the collection of all correlation coefficients in $R_k$. Clearly $\hat{\beta}$ depends on $\rho$ and $\phi$, which also needs to be estimated. If we define the Pearson residual $e_{kj} = (y_{kj} - \mu_{kj})/\sqrt{v(\mu_{kj})}$, then $\hat{\phi} = \frac{1}{(\Sigma_{k=1}^K J_k) - p} \sum_{k=1}^K \sum_{j=1}^{J_k} e_{kj}^2$. Next, $\hat{\rho}$ is estimated as a function of $\phi$ and $e_{kj}$. The exact formula of $\hat{\rho}$ depends on the correlation structure $R$, and a few examples of $\hat{\rho}$ under different structures are given in Liang et al [27] and Wang [33]. Because the Pearson residuals $e_{kj}$'s also depend on $\hat{\beta}$, it yields an iterative scheme which switches between estimating $\beta$ from fixed value of $\hat{\phi}$ and $\hat{\rho}$ and estimating $\phi$ and $\rho$ for a fixed value of $\hat{\beta}$. Under GEE theory [27], this scheme yields a consistent estimate for $\beta$. Moreover $\hat{\beta}$ is asymptotically normally distributed with mean $\beta$ and variance

$$V_\beta = (\Sigma_{k=1}^K D_k' V_k^{-1} D_k)^{-1} \{\Sigma_{k=1}^K D_k' V_k^{-1} \text{Cov}(y_k) V_k^{-1} D_k\} (\Sigma_{k=1}^K D_k' V_k^{-1} D_k)^{-1} \tag{3}$$

where $\text{Cov}(y_k)$ is the true underlying covariance matrix of $y_k$. The consistent estimator of $V_\beta$, $\hat{V}_\beta$, is achieved by replacing $\hat{\beta}, \hat{\rho}, \hat{\phi}$ and $\{y_k - \mu_k(\hat{\beta})\}\{y_k - \mu_k(\hat{\beta})\}'$ for $\beta, \rho, \phi$ and $\text{Cov}(y_k)$.

GEE method yields consistent estimator of $\beta$, even if the structure of working correlation matrix is not correctly specified. The misspecified $R_k(\rho)$ only affects the efficiency of $\hat{\beta}$. The consistent estimation of correlation matrix $R_k(\hat{\rho})$, however, relies on correct specification of the correlation structure.

For testing a hypothesis of $H_0$: $C\beta = c$, a Wald test statistic can be used with the form

$$W = (C\hat{\beta} - c)'(C\hat{V}_\beta C')^{-1}(C\hat{\beta} - c) \tag{4}$$

and $W \xrightarrow{d} \chi^2_{(q)}$, where $q$ is the rank of matrix $C$.

**Estimating predictors effects on OTUs.** Based on the GEE framework, we develop the MTLC model to assess the association between OTUs and the predictors of interest, accounting for the correlation of repeated OTU measures. To deal with the excess zeros of OTUs using MTLC model, first we convert quantitative OTU observations to binary outcomes (0 and 1), indicating the prevalence of OTU in each observation. Next, we focus on the OTU relative abundance (RA) of each non-zero observation, and assume the RAs following normal distribution after log transformation. We use two separate GEE models, one for assessing the predictor effects on OTU prevalence, and the other for assessing the predictor effects on positive RA. The predictors' overall effects are finally tested by combining the test statistics from these two GEE models.

Formally, for $k = 1, \ldots K$ and $j = 1, \ldots, J_k$, we assume each OTU observation $y_{kj}$ follows a mixture of Bernoulli and log-normal distribution: suppose $y_{kj}^{(0)}$ follows a Bernoulli distribution with $P(y_{kj}^{(0)} = 1) = \mu_{kj}^{(0)}$, and $y_{kj}^{(+)}$ follows a normal distribution such that $y_{kj}^{(+)} \sim N(\mu_{kj}^{(+)}, \sigma^2)$, then the distribution function of $y_{kj}$ is

$$F(y) = \begin{cases} 1 - \mu_{kj}^{(0)} & y = 0 \\ 1 - \mu_{kj}^{(0)} + \mu_{kj}^{(0)}\Phi(\log_{10} y) & y > 0 \end{cases}$$

where $\Phi$ is the distribution function of $y_{kj}^{(+)}$. By definition, $y_{kj}^{(0)}$ represents OTU prevalence

observations because

$$y_{kj}^{(0)} = \begin{cases} 0 & y_{kj} = 0 \\ 1 & y_{kj} > 0 \end{cases}$$

and $y_{kj}^{(+)}$ represents the positive RAs because $\log_{10} y_{kj} = y_{kj}^{(+)}$ for all $y_{kj} > 0$. We use $\boldsymbol{y}_k^{(0)}$ to denote the vector of all $y_{kj}^{(0)}$, and $\boldsymbol{y}_k^{(+)}$ to denote the the subset of $y_{kj}^{(+)}$ where $y_{kj} > 0$.

Rather than running generalized linear model directly on $\boldsymbol{y}_k$, we apply GEE method separately on $\boldsymbol{y}_k^{(0)}$ and $\boldsymbol{y}_k^{(+)}$. For these two GEE models, the predictors' design matrices $\boldsymbol{X}_k$ do not have to be the same in principal, although they could be the same in many practical situations. Without loss of generality we simply assume the predictors are same in each part of the GEE model in this paper. We choose logit link function for binary outcomes and identity link function for log transformed non-zero outcomes, and the two parts of the GEE model are

$$\log\left(\frac{\mu_{kj}^{(0)}}{1 - \mu_{kj}^{(0)}}\right) = \boldsymbol{x}_{kj}'\boldsymbol{\beta}^{(0)} \tag{5}$$

and

$$\mu_{kj}^{(+)} = \boldsymbol{x}_{kj}'\boldsymbol{\beta}^{(+)} \tag{6}$$

Using iterative scheme discussed in section "Generalized estimating equation framework" on $\boldsymbol{y}_k^{(0)}$ and $\boldsymbol{y}_k^{(+)}$, we can achieve the corresponding parameter estimation $\hat{\boldsymbol{\beta}}^{(0)}$ and $\hat{\boldsymbol{\beta}}^{(+)}$.

**Hypothesis testing.** For testing if the predictors have effects to either the prevalence of OTUs or the quantitative amount of RA, the null hypothesis is

$$H_0 : \boldsymbol{C}^{(0)}\boldsymbol{\beta}^{(0)} = \boldsymbol{c}^{(0)} \text{ and } \boldsymbol{C}^{(+)}\boldsymbol{\beta}^{(+)} = \boldsymbol{c}^{(+)}.$$

Assuming same $\boldsymbol{X}_k$ for the $\boldsymbol{y}_k^{(0)}$ part and $\boldsymbol{y}_k^{(+)}$ part of GEE model, $\boldsymbol{\beta}^{(0)}$ and $\boldsymbol{\beta}^{(+)}$ will have the same dimension $p$. Moreover, $\boldsymbol{C}^{(0)} = \boldsymbol{C}^{(+)}$ and $\boldsymbol{c}^{(0)} = \boldsymbol{c}^{(+)}$ in many practical situations. For example, if we want to test the first $q$ predictors in $\boldsymbol{X}_k$ and the rest $p - q$ extra covariates are not of interest, then

$$\boldsymbol{C}^{(0)} = \boldsymbol{C}^{(+)} = \begin{pmatrix} \boldsymbol{I}_{q \times q} & \boldsymbol{0}_{q \times (p-q)} \\ \boldsymbol{0}_{(p-q) \times q} & \boldsymbol{0}_{(p-q) \times (p-q)} \end{pmatrix}, \boldsymbol{c}^{(0)} = \boldsymbol{c}^{(+)} = \boldsymbol{0}.$$

For each part of $H_0$, the corresponding test statistics $W^{(0)}$ and $W^{(+)}$ are computed following Eq 4.

It follows section "Generalized estimating equation framework" that $W^{(0)} \xrightarrow{d} \chi^2_{(q^{(0)})}$ and $W^{(+)} \xrightarrow{d} \chi^2_{(q^{(+)})}$. Besides, for jointly testing two null hypotheses by the combined test on $W^{(0)}$ and $W^{(+)}$, we adopt Cauchy combination test [35], which does not require the independence assumption between $W^{(0)}$ and $W^{(+)}$. Let $p^{(0)}$ and $p^{(+)}$ be the corresponding p-values, then the Cauchy combination test statistic is

$$W_{MTLC} = 0.5 tan[(0.5 - p^{(0)})\pi] + 0.5 tan[(0.5 - p^{(+)})\pi] \xrightarrow{d} Cauchy(0, 1) \tag{7}$$

**Estimating correlation coefficients.** In our proposed MTLC model, the correlation structure is based on OTU taxonomic structure and characterizing correlations between repeated

measures. Here we assume the two GEE models corresponding to the OTU prevalence part and positive RA part have the same correlation structure $R$. However, the estimated values of correlation coefficients, $\hat{\rho}^{(0)}$ and $\hat{\rho}^{(+)}$, may be different for each part of the GEE model. For $y_k^{(0)}$ and $y_k^{(+)}$, $\hat{\rho}^{(0)}$ and $\hat{\rho}^{(+)}$ are estimated separately following the iterative scheme discussed in section "Generalized estimating equation framework".

It needs to be noted that GEE models do not require each cluster has equal cluster size, which could happen, for example, in unbalanced study designs and/or when some observations are missing. Even if $y_k^{(0)}$ has equal size for all $k$, $y_k^{(+)}$ may have different sizes as it is a collection of only positive RAs. It implies that the dimension of $R$ may be greater than the length of $y_k^{(0)}$ and $y_k^{(+)}$ for some $k$. In such case, the rows and columns in $R$ corresponding to empty values of OTU observations need to be removed, and we denote the modified correlation structure matrices by $R_k^{(0)}(\rho)$ and $R_k^{(+)}(\rho)$ correspondingly for each $k$. When applying the estimating equations in our MTLC model, we essentially use $R_k^{(0)}(\rho)$ and $R_k^{(+)}(\rho)$ as the working correlation matrices.

## Simulation

### Simulation settings

Simulation studies are designed to simulate zero inflated multivariate normal distribution to reflect the correlation of $-\log_{10}$ transformed OTUs. To achieve this, we simulate both multivariate Bernoulli distribution samples $Y^{(0)}$ and truncated multivariate normal distribution samples $Z$ of size $K$ and length $J$. Multivariate normal distributions are truncated to generate positive samples because all $-\log_{10}$ transformed RAs should be positive. We further assume a single binary predictor $X$, where $X$ also has dimension $K \times J$, and the mean of $Y^{(0)}$ and $Z$ depend on $X$. Specifically, we simulate $Y^{(0)} \sim Bernoulli_J(\frac{exp(X\beta^{(0)})}{1+exp(X\beta^{(0)})})$, and $Z \sim N_J(X\beta^{(+)}, R)$ truncated at 0. The zero-inflated multivariate normal distribution samples are computed as $Y = Y^{(0)} Z$. $Y$ is indirectly associated with $X$ via $Y^{(0)}$ and $Z$.

For illustration purpose, we assume the simplest correlation structure, i.e., two correlated OTUs under taxonomic structure and two repeated measures at different time points). The correlation matrix $R$ is then derived following section "Incorporating taxonomic structure with repeated measures":

$$R = \begin{pmatrix} \rho_{(\mathbb{D},\mathbb{D})} & \rho_{(\mathbb{D},\hat{\imath})} & \rho_{(\mathbb{I},\mathbb{D})} & \rho_{(\mathbb{I},\hat{\imath})} \\ \rho_{(\mathbb{D},\hat{\imath})} & \rho_{(\mathbb{D},\mathbb{D})} & \rho_{(\mathbb{I},\hat{\imath})} & \rho_{(\mathbb{I},\mathbb{D})} \\ \rho_{(\mathbb{I},\mathbb{D})} & \rho_{(\mathbb{I},\hat{\imath})} & \rho_{(\mathbb{D},\mathbb{D})} & \rho_{(\mathbb{D},\hat{\imath})} \\ \rho_{(\mathbb{I},\hat{\imath})} & \rho_{(\mathbb{I},\mathbb{D})} & \rho_{(\mathbb{D},\hat{\imath})} & \rho_{(\mathbb{D},\mathbb{D})} \end{pmatrix}.$$

$\rho_{(\mathbb{D},\mathbb{D})} = 1$, $\rho_{(\mathbb{D},\hat{\imath})}$ and $\rho_{(\mathbb{I},\mathbb{D})}$ denote the correlation between two time points and between two OTUs. $\rho_{(\mathbb{I},\hat{\imath})}$ represents the correlation of observations from different OTU and different time points, which is not of primary interest. We assume the simulated multivariate Bernoulli and multivariate normal distribution follow the same correlation structure $R$, but the correlation coefficients $\hat{\rho}^{(0)}$ and $\hat{\rho}^{(+)}$ can be different.

After achieving the zero-inflated multivariate normal distribution samples $Y$, we run a GEE logistic model following Eq 5 to estimate the effects of $X$ to OTU prevalence $Y^{(0)}$, and GEE linear model following Eq 6 to estimate $X$ effects to the non-zero RAs $Y^{(+)}$, where $Y^{(+)}$ is the subset of $Z$ such that $y_{kj}^{(+)} = z_{kj}|(y_{kj}^{(0)} = 1)$. Under GEE theory, both $Y^{(0)}$ and $Y^{(+)}$ yield consistent

estimations of $\beta$ and $\rho$. However, we simulate $Z$ rather than $Y^{(+)}$, where $Z$ and $Y^{(+)}$ may not yield same estimations in general. To solve this issue, we simulate $Z$ and $Y^{(0)}$ independently, which implies that $y_{kj}^{(+)}$ has the same distribution as $z_{kj}$. Therefore, $Z$ also yields consistent estimations of $\beta$ and $\rho$.

Different from some literature that $Y$ is directly simulated, we conducted our stimulation on $Y^{(0)}$ and $Z$ separately. This is because following the mixture distribution framework, we conduct two separate GEE models on $Y^{(0)}$ and $Y^{(+)}$ rather than one model directly on $Y$. In this way, we can clearly specify the true values of predictor's main effects and OTU correlations in simulation settings, and evaluate if the estimations of these values are unbiased explicitly. As a sensitivity analysis to evaluate the robustness of our model performance, we also simulate $Y^{(0)}$ and $Z$ from (generalized) linear mixed model. Results are presented in S1 Appendix.

## Inferences for predictor's main effects

First, we evaluate the performance of our proposed MTLC model for estimating and testing the main effects or the predictor $X$. Let $\beta^{(0)}$ denote the effects on OTU prevalence and $\beta^{(+)}$ denote the effects on the $\log_{10}$ transformed none-zero RA. We evaluate the unbiasedness of estimated $\hat{\beta}^{(0)}$, $\hat{\beta}^{(+)}$, Type I error for testing $\beta^{(0)} = \beta^{(+)} = 0$ and test power when $\beta^{(0)}$ and/or $\beta^{(+)} \neq 0$. OTU observations are simulated under the simulation settings discussed in section "Simulation settings" with sample size $K = 1000$ and various combinations of $\beta^{(0)}$ and $\beta^{(+)}$ values. We assume $\rho_{(\mathbb{D},\mathbb{I})} = \rho_{(\mathbb{I},\mathbb{D})} = 0.3$ and $\rho_{(\mathbb{I},\mathbb{I})} = 0$ for both the multivariate normal and multivariate Bernoulli distribution. $\beta$, Type I errors and powers are estimated based on 1000 replications. The computation time is about 4 hours to complete all 1000 replications on a desktop computer with quad-core processor and 8GB of RAM.

Next we compare our MTLC model to other models. All models are described in Table 1.

For each model, the estimated $\hat{\beta}^{(0)}$, $\hat{\beta}^{(+)}$, Type I error and power are summarized in Table 2. We find all estimates of $\beta^{(0)}$ and $\beta^{(+)}$ are unbiased under MTLC model. For the one-part models, because there is no true value of $\beta$ as a mixture of $\beta^{(0)}$ and $\beta^{(+)}$, the unbiasedness of estimated $\beta$ cannot be evaluated. Regarding the variations of estimated $\hat{\beta}$, the 2.5 and 97.5 percentile of the empirical distributions of $\hat{\beta}$ are shown in S1 Appendix.

Given the true Type I error at 0.05, 2P_ind and 1P_ind model have inflated Type I error, and all other estimated Type I errors are accurate. It needs to be noted that when only one of $\beta^{(0)}$ and $\beta^{(+)}$ equal to 0, the Type I error estimation is still accurate. For example, when

**Table 1. Description of each model compared by simulation study.**

| Name | Formula | Description |
|---|---|---|
| GEE$^{(0)}$ | $Y^{(0)} \overset{GEE}{\sim} X$ | The logistic regression part of GEE for OTU prevalence |
| GEE$^{(+)}$ | $Y^{(+)} \overset{GEE}{\sim} X$ | The linear regression part of GEE for non-zero RAs |
| MTLC | $Y^{(0)} \overset{GEE}{\sim} X$ $Y^{(+)} \overset{GEE}{\sim} X$ | two-part GEE: our proposed microbiome taxonomic longitudinal correlation model model for OTU prevalence, linear model for non-zero RAs |
| 2P_ind | $Y^{(0)} \sim X$ $Y^{(+)} \sim X$ | two-part independence: assuming no correlation, logistic |
| 1P_GEE | $Y \overset{GEE}{\sim} X$ | one-part GEE: assuming same correlation structure, but only one GEE linear model for all 0 and non-zero RAs |
| 1P_ind | $Y \sim X$ | one-part independence: assuming no correlation and only one simple linear model for all 0 and non-zero RAs |
| 1P_RE | $Y \sim X + \gamma_1 + \gamma_2$ | one-part linear mixed model with random intercepts: $\gamma_1, \gamma_2$ represents random intercepts of time points and OTUs |

**Table 2. Estimated $\hat{\beta}$, Type I error and power, from 1000 replications.**

| $(\beta^{(0)}, \beta^{(+)})$ | Estimates | GEE$^{(0)}$ | GEE$^{(+)}$ | MTLC | 2P_ind | 1P_GEE | 1P_ind | 1P_RE |
|---|---|---|---|---|---|---|---|---|
| (0,0) | $\hat{\beta}$ | NA | NA | NA | NA | 0.000 | 0.000 | 0.000 |
| | $\hat{\beta}^{(0)}$ | 0.001 | NA | 0.001 | 0.001 | NA | NA | NA |
| | $\hat{\beta}^{(+)}$ | NA | 0.000 | 0.000 | 0.000 | NA | NA | NA |
| | T1E | 0.056 | 0.038 | 0.039 | 0.120 | 0.050 | 0.116 | 0.047 |
| (0,0.05) | $\hat{\beta}$ | NA | NA | NA | NA | 0.027 | 0.027 | 0.027 |
| | $\hat{\beta}^{(0)}$ | 0.002 | NA | 0.002 | 0.002 | NA | NA | NA |
| | $\hat{\beta}^{(+)}$ | NA | 0.052 | 0.052 | 0.052 | NA | NA | NA |
| | Power | 0.045 | 0.512 | 0.421 | 0.583 | 0.201 | 0.332 | 0.199 |
| (0,-0.05) | $\hat{\beta}$ | NA | NA | NA | NA | -0.026 | -0.026 | -0.026 |
| | $\hat{\beta}^{(0)}$ | -0.001 | NA | -0.001 | -0.001 | NA | NA | NA |
| | $\hat{\beta}^{(+)}$ | NA | -0.050 | -0.050 | -0.050 | NA | NA | NA |
| | Power | 0.048 | 0.487 | 0.394 | 0.552 | 0.187 | 0.312 | 0.188 |
| (0.1,0) | $\hat{\beta}$ | NA | NA | NA | NA | 0.051 | 0.051 | 0.051 |
| | $\hat{\beta}^{(0)}$ | 0.101 | NA | 0.101 | 0.101 | NA | NA | NA |
| | $\hat{\beta}^{(+)}$ | NA | 0.001 | 0.001 | 0.001 | NA | NA | NA |
| | Power | 0.693 | 0.050 | 0.609 | 0.772 | 0.571 | 0.712 | 0.570 |
| (0.1,0.05) | $\hat{\beta}$ | NA | NA | NA | NA | 0.075 | 0.075 | 0.075 |
| | $\hat{\beta}^{(0)}$ | 0.100 | NA | 0.100 | 0.100 | NA | NA | NA |
| | $\hat{\beta}^{(+)}$ | NA | 0.049 | 0.049 | 0.049 | NA | NA | NA |
| | Power | 0.705 | 0.487 | 0.771 | 0.887 | 0.862 | 0.934 | 0.866 |
| (0.1,-0.05) | $\hat{\beta}$ | NA | NA | NA | NA | 0.025 | 0.025 | 0.025 |
| | $\hat{\beta}^{(0)}$ | 0.099 | NA | 0.099 | 0.099 | NA | NA | NA |
| | $\hat{\beta}^{(+)}$ | NA | -0.050 | -0.050 | -0.049 | NA | NA | NA |
| | Power | 0.696 | 0.481 | 0.800 | 0.896 | 0.171 | 0.287 | 0.171 |
| (-0.1,0) | $\hat{\beta}$ | NA | NA | NA | NA | -0.051 | -0.051 | -0.051 |
| | $\hat{\beta}^{(0)}$ | -0.101 | NA | -0.101 | -0.101 | NA | NA | NA |
| | $\hat{\beta}^{(+)}$ | NA | -0.001 | -0.001 | -0.001 | NA | NA | NA |
| | Power | 0.700 | 0.054 | 0.612 | 0.781 | 0.575 | 0.698 | 0.571 |
| (-0.1,0.05) | $\hat{\beta}$ | NA | NA | NA | NA | -0.026 | -0.026 | -0.026 |
| | $\hat{\beta}^{(0)}$ | -0.102 | NA | -0.102 | -0.102 | NA | NA | NA |
| | $\hat{\beta}^{(+)}$ | NA | 0.050 | 0.050 | 0.050 | NA | NA | NA |
| | Power | 0.719 | 0.483 | 0.803 | 0.905 | 0.188 | 0.304 | 0.183 |
| (-0.1,-0.05) | $\hat{\beta}$ | NA | NA | NA | NA | -0.075 | -0.075 | -0.075 |
| | $\hat{\beta}^{(0)}$ | -0.099 | NA | -0.099 | -0.099 | NA | NA | NA |
| | $\hat{\beta}^{(+)}$ | NA | -0.050 | -0.050 | -0.050 | NA | NA | NA |
| | Power | 0.694 | 0.471 | 0.786 | 0.906 | 0.887 | 0.949 | 0.887 |

$(\beta^{(0)}, \beta^{(+)}) = (0, 0.05)$, the GEE$^{(0)}$ model for testing $\beta^{(0)} = 0$ has Type I error 0.062, which is not affected by the non-zero value of $\beta^{(+)}$. It further confirms the independence of the linear and logistic regression parts in the two-part model.

We also evaluate the power performance of different models. The power of 2P_ind and 1P_ind model are inflated due to Type I error inflation. Our proposed MTLC model is most

powerful in general. When one of $\beta^{(0)}$ and $\beta^{(+)}$ is 0, the MTLC model is slightly less powerful than one of GEE$^{(0)}$ and GEE$^{(+)}$ model which only tests the part that $\beta \neq 0$. However, when both $\beta^{(0)}$ and $\beta^{(+)}$ are non-zero, the MTLC model is much more powerful than both GEE$^{(0)}$ and GEE$^{(+)}$ model. The 1P_GEE model and 1P_RE model have similar powers. It needs to be noted that the 1P_RE model is not able to accommodate negative correlations due to the natural or random effects. This is the reason that we choose $\rho_{01}$ and $\rho_{10}$ to be positive in the simulation settings. When the true correlations are negative, the 1P_RE model simply reduces to 1P_ind model. Comparing to the MTLC model, the power of the one-part models drops dramatically when $\beta^{(0)}$ and $\beta^{(+)}$ have opposite sign. This is because the positive effect cancels out the negative effects in one-part models, but both effects are well captured in two-part models. When $\beta^{(0)}$ and $\beta^{(+)}$ have same direction, we do observe some cases that the power of one-part models are larger. This is related to how to deal with the excess zeros in the one-part models. Detailed discussion about this issue is provided in section "Two-part vs. one-part models".

## Estimations for the correlation coefficients

The MTLC model can also provide estimations of correlation coefficients. First we evaluate the unbiasedness of the correlation estimates. Let $\boldsymbol{\rho^{(0)}}$ and $\boldsymbol{\rho^{(+)}}$ be correlation coefficients in GEE$^{(0)}$ and GEE$^{(+)}$ model. In simulation settings, we choose $\rho_{(\mathbb{D},\mathbb{i})}^{(0)} = \rho_{(\mathbb{i},\mathbb{D})}^{(0)} = 0.5$ and $\rho_{(\mathbb{D},\mathbb{i})}^{(+)} = \rho_{(\mathbb{i},\mathbb{D})}^{(+)} = -0.3$, $\beta^{(0)} = -0.1$ and $\beta^{(+)} = 0.05$. The specified $\beta$ values do not affect the estimation of $\boldsymbol{\rho}$. Sample size $K = 1000$ and number of replications remains to be 1000.

The correlation structure of OTUs is based on the taxonomic structure, which is usually known in practice. However, the correlation structure of repeated measures within each OTU may not be known and usually requires subjective assumptions. One merit of GEE model is that even if the assumption of correlation structure is not correct, it does not affect the estimation of main effect $\beta$. The $\hat{\beta}$ estimations are consistent under different assumptions of correlation structure, as illustrated by Yan [36] and confirmed by our simulation study (results not shown). Besides that, we evaluate the consistency of correlation estimations under wrong correlative structure setting.

In contrast to the correct correlation structure $\boldsymbol{R}$, we first construct a model with a correlation matrix assuming that OTUs are independent while time points are still correlated. After that, we construct another model with correlation matrix assuming that time points are independent while OTUs are still correlated. When OTUs are assumed to be independent, the GEE model may only estimate $\rho_{(\mathbb{D},\mathbb{i})}$; when time points are independent, the GEE model may only estimate $\rho_{(\mathbb{i},\mathbb{D})}$. The correlation estimations are summarized in Table 3.

From Table 3, the correlation estimates under true correlation structure are all unbiased. When the correlation structure is not correctly specified, it may not estimate all correlation

**Table 3. Estimated GEE correlations under correct correlation structure, OTU independence structure and time points independence structure, compared to Pearson correlations.**

| Cor | True | Pearson | True structure | OTU ind | Time points ind |
|---|---|---|---|---|---|
| $\rho_{(\mathbb{D},\mathbb{i})}^{(0)}$ | 0.5 | 0.497 | 0.495 | 0.495 | NA |
| $\rho_{(\mathbb{i},\mathbb{D})}^{(0)}$ | 0.5 | 0.498 | 0.496 | NA | 0.496 |
| $\rho_{(\mathbb{i},\mathbb{i})}^{(0)}$ | 0 | 0.000 | -0.002 | NA | NA |
| $\rho_{(\mathbb{D},\mathbb{i})}^{(+)}$ | -0.3 | -0.295 | -0.299 | -0.300 | NA |
| $\rho_{(\mathbb{i},\mathbb{D})}^{(+)}$ | -0.3 | -0.296 | -0.299 | NA | -0.299 |
| $\rho_{(\mathbb{i},\mathbb{i})}^{(+)}$ | 0 | -0.001 | -0.001 | NA | NA |

coefficients for the correct correlation structure, but more interestingly, for those correlation coefficients which can be estimated under the misspecified structure, the estimation remains to be unbiased. It implies that if we are not interested in estimating all correlations in the correct correlation structure, we can simplify the correlation structure. For example, because the estimation of $\rho_{(\mathbb{I},\mathbb{I})}$ is not of interest, we can set it to 0 without affecting the estimation of $\rho_{(\mathbb{D},\mathbb{I})}$ and $\rho_{(\mathbb{I},\mathbb{D})}$.

The correlation structure only contains two OTUs and two time points, so the GEE correlation estimates are essentially pairwise correlations, and thus they can be compared with corresponding Pearson correlation coefficients. Both results are consistent as expected. The merit of our MTLC model is that when the correlation structure is more complicated and the pairwise Pearson correlation is not available, it may still provide unbiased estimation of the correlation matrix.

## Two-part vs. one-part models

For one-part models, if we take $-\log_{10}$ transformation of both the non-zero RAs and 0, then all 0 becomes $\infty$. To solve this issue, one common approach is to change all 0 to some small value close to 0, such as $10^{-5}$. However, we find the one-part model test powers are sensitive to this arbitrary small value. In Table 4, we replace $-\log_{10} 0$ by 6, 5 4 and 3 and compare corresponding test powers with the MTLC model. We only present the 1P_GEE model as we have shown in Table 2 that the 1P_RE model has similar power to 1P_GEE.

Table 4 indicates that there is no optimal choice of the value for replacing 0 RAs. For each value selected, depending on $(\beta^{(0)}, \beta^{(+)})$, there may exist some situations such that the one-part model has comparable power or even slightly better power than corresponding two-part model (e.g., 0.650 vs. 0.609 when $(\beta^{(0)}, \beta^{(+)}) = (0.1, 0)$ and replacing 0 by $10^{-6}$), but the power loss is much more significant for some other values of $\beta$ (e.g., 0.138 vs. 0.421 when $(\beta^{(0)}, \beta^{(+)}) = (0, 0.05)$ and replacing 0 by $10^{-6}$). We conclude that our MTLC models has superior and robust power performance compared to the one-part models, and suggest readers avoid using the one-part models in practice when there are excessive numbers of 0s in OTU data.

## Application

We implement our proposed MTLC model on a twin study described in Turnbaugh et al. [37]. The full dataset is provided in the supporting information S1 Data. The data consists of 54 families and each family has a pair of twins. Each individual has at most two observations at two time points. The primary research question is to assess the association between obesity status (lean, overweight or obese) and OTUs, and estimate the correlations between two time

**Table 4. Comparing test powers from 1P_GEE model to MTLC model when $-\log_{10} 0$ are replaced by 6, 5 4 and 3.**

| $(\beta^{(0)}, \beta^{(+)})$ | MTLC | $-\log_{10} 0 = 6$ | $-\log_{10} 0 = 5$ | $-\log_{10} 0 = 4$ | $-\log_{10} 0 = 3$ |
|---|---|---|---|---|---|
| (0,0) | 0.039 | 0.038 | 0.052 | 0.040 | 0.044 |
| (0,0.05) | 0.421 | 0.138 | 0.156 | 0.304 | 0.478 |
| (0,-0.05) | 0.394 | 0.122 | 0.176 | 0.284 | 0.468 |
| (0.1,0) | 0.609 | 0.650 | 0.528 | 0.308 | 0.040 |
| (0.1,0.05) | 0.771 | 0.890 | 0.888 | 0.864 | 0.456 |
| (0.1,-0.05) | 0.764 | 0.346 | 0.218 | 0.050 | 0.484 |
| (-0.1,0) | 0.612 | 0.660 | 0.576 | 0.340 | 0.050 |
| (-0.1,0.05) | 0.803 | 0.306 | 0.166 | 0.052 | 0.486 |
| (-0.1,-0.05) | 0.786 | 0.846 | 0.854 | 0.844 | 0.472 |

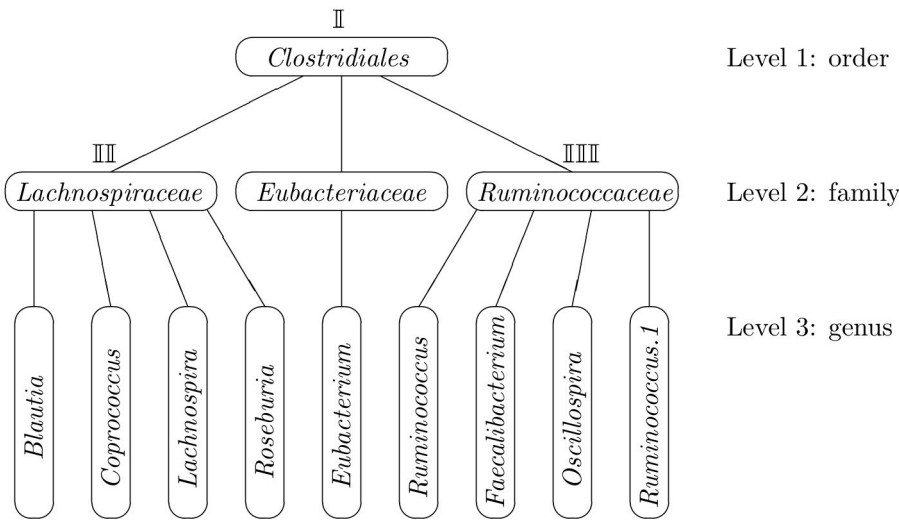

**Fig 3. Taxonomic structure of 9 OTUs.**

points, each pair of twins and OTUs. For illustration purpose, we only analyze OTUs within the order *Clostridiales*, which consists of 9 OTUs at genus level. The taxonomic structure of these 9 OTUs are shown in Fig 3.

From Fig 3, all 9 OTUs begin to belong to the same taxa (*Clostridiales*) at level order, and each of the 9 OTUs belongs to a different taxon at level genus. We define level order as level 1, level family as level 2 and level genus as level 3, thus $I = 3$. Accordingly, the numerical representation of the taxonomic structure is $n_1 = 9$, $n_2 = (4, 1, 4)$, $n_3 = (1, 1, 1, 1, 1, 1, 1, 1, 1)$.

Next, following the 4 steps described in section "Taxonomic structure of OTUs", the taxonomic structure matrix is

$$\boldsymbol{\Gamma} = \begin{pmatrix} \mathbb{D} & \mathbb{I} & \mathbb{I} & \mathbb{I} & \mathbb{III} & \mathbb{III} & \mathbb{III} & \mathbb{III} & \mathbb{III} \\ \mathbb{I} & \mathbb{D} & \mathbb{I} & \mathbb{I} & \mathbb{III} & \mathbb{III} & \mathbb{III} & \mathbb{III} & \mathbb{III} \\ \mathbb{I} & \mathbb{I} & \mathbb{D} & \mathbb{I} & \mathbb{III} & \mathbb{III} & \mathbb{III} & \mathbb{III} & \mathbb{III} \\ \mathbb{I} & \mathbb{I} & \mathbb{I} & \mathbb{D} & \mathbb{III} & \mathbb{III} & \mathbb{III} & \mathbb{III} & \mathbb{III} \\ \mathbb{III} & \mathbb{III} & \mathbb{III} & \mathbb{III} & \mathbb{D} & \mathbb{III} & \mathbb{III} & \mathbb{III} & \mathbb{III} \\ \mathbb{III} & \mathbb{III} & \mathbb{III} & \mathbb{III} & \mathbb{III} & \mathbb{D} & \mathbb{II} & \mathbb{II} & \mathbb{II} \\ \mathbb{III} & \mathbb{III} & \mathbb{III} & \mathbb{III} & \mathbb{III} & \mathbb{II} & \mathbb{D} & \mathbb{II} & \mathbb{II} \\ \mathbb{III} & \mathbb{III} & \mathbb{III} & \mathbb{III} & \mathbb{III} & \mathbb{II} & \mathbb{II} & \mathbb{D} & \mathbb{II} \\ \mathbb{III} & \mathbb{III} & \mathbb{III} & \mathbb{III} & \mathbb{III} & \mathbb{II} & \mathbb{II} & \mathbb{II} & \mathbb{D} \end{pmatrix}.$$

Because each OTU is observed at two time points for a pair of twins, the repeated measure correlation structure following section "Modelling correlations from repeated measures" is

$$\boldsymbol{\Omega} = \begin{pmatrix} \mathbb{D} & \hat{\mathbb{i}} & \hat{\mathbb{i}}\hat{\mathbb{i}} & \hat{\mathbb{i}}\hat{\mathbb{i}}\hat{\mathbb{i}} \\ \hat{\mathbb{i}} & \mathbb{D} & \hat{\mathbb{i}}\hat{\mathbb{i}}\hat{\mathbb{i}} & \hat{\mathbb{i}}\hat{\mathbb{i}} \\ \hat{\mathbb{i}}\hat{\mathbb{i}} & \hat{\mathbb{i}}\hat{\mathbb{i}}\hat{\mathbb{i}} & \mathbb{D} & \hat{\mathbb{i}} \\ \hat{\mathbb{i}}\hat{\mathbb{i}}\hat{\mathbb{i}} & \hat{\mathbb{i}}\hat{\mathbb{i}} & \hat{\mathbb{i}} & \mathbb{D} \end{pmatrix}.$$

The dimension of $\boldsymbol{\Gamma}$ and $\boldsymbol{\Omega}$ are $N = 9$ and $L = 4$, so as described in section "Incorporating taxonomic structure with repeated measures", the integrative correlation matrix $\boldsymbol{R}$ has

dimension $J = N \times L = 36$. For $a = 1, \ldots, 9$ and $b = 1, \ldots, 9$, if $\Gamma_{ab} = \mathbb{D}$, then

$$\mathbf{\Omega}^{ab} = \mathbf{\Omega}(\mathbb{D}) = \begin{pmatrix} \rho_{(\mathbb{D},\mathbb{D})} & \rho_{(\mathbb{D},\mathring{\mathbb{I}})} & \rho_{(\mathbb{D},\mathring{\mathbb{I}}\mathring{\mathbb{I}})} & \rho_{(\mathbb{D},\mathring{\mathbb{I}}\mathring{\mathbb{I}}\mathring{\mathbb{I}})} \\ \rho_{(\mathbb{D},\mathring{\mathbb{I}})} & \rho_{(\mathbb{D},\mathbb{D})} & \rho_{(\mathbb{D},\mathring{\mathbb{I}}\mathring{\mathbb{I}}\mathring{\mathbb{I}})} & \rho_{(\mathbb{D},\mathring{\mathbb{I}}\mathring{\mathbb{I}})} \\ \rho_{(\mathbb{D},\mathring{\mathbb{I}}\mathring{\mathbb{I}})} & \rho_{(\mathbb{D},\mathring{\mathbb{I}}\mathring{\mathbb{I}}\mathring{\mathbb{I}})} & \rho_{(\mathbb{D},\mathbb{D})} & \rho_{(\mathbb{D},\mathring{\mathbb{I}})} \\ \rho_{(\mathbb{D},\mathring{\mathbb{I}}\mathring{\mathbb{I}}\mathring{\mathbb{I}})} & \rho_{(\mathbb{D},\mathring{\mathbb{I}}\mathring{\mathbb{I}})} & \rho_{(\mathbb{D},\mathring{\mathbb{I}})} & \rho_{(\mathbb{D},\mathbb{D})} \end{pmatrix};$$

if $\Gamma_{ab} = \mathbb{I}$, then

$$\mathbf{\Omega}^{ab} = \mathbf{\Omega}(\mathbb{I}) = \begin{pmatrix} \rho_{(\mathbb{I},\mathbb{D})} & \rho_{(\mathbb{I},\mathring{\mathbb{I}})} & \rho_{(\mathbb{I},\mathring{\mathbb{I}}\mathring{\mathbb{I}})} & \rho_{(\mathbb{I},\mathring{\mathbb{I}}\mathring{\mathbb{I}}\mathring{\mathbb{I}})} \\ \rho_{(\mathbb{I},\mathring{\mathbb{I}})} & \rho_{(\mathbb{I},\mathbb{D})} & \rho_{(\mathbb{I},\mathring{\mathbb{I}}\mathring{\mathbb{I}}\mathring{\mathbb{I}})} & \rho_{(\mathbb{I},\mathring{\mathbb{I}}\mathring{\mathbb{I}})} \\ \rho_{(\mathbb{I},\mathring{\mathbb{I}}\mathring{\mathbb{I}})} & \rho_{(\mathbb{I},\mathring{\mathbb{I}}\mathring{\mathbb{I}}\mathring{\mathbb{I}})} & \rho_{(\mathbb{I},\mathbb{D})} & \rho_{(\mathbb{I},\mathring{\mathbb{I}})} \\ \rho_{(\mathbb{I},\mathring{\mathbb{I}}\mathring{\mathbb{I}}\mathring{\mathbb{I}})} & \rho_{(\mathbb{I},\mathring{\mathbb{I}}\mathring{\mathbb{I}})} & \rho_{(\mathbb{I},\mathring{\mathbb{I}})} & \rho_{(\mathbb{I},\mathbb{D})} \end{pmatrix};$$

if $\Gamma_{ab} = \mathbb{II}$, then

$$\mathbf{\Omega}^{ab} = \mathbf{\Omega}(\mathbb{II}) = \begin{pmatrix} \rho_{(\mathbb{II},\mathbb{D})} & \rho_{(\mathbb{II},\mathring{\mathbb{I}})} & \rho_{(\mathbb{II},\mathring{\mathbb{I}}\mathring{\mathbb{I}})} & \rho_{(\mathbb{II},\mathring{\mathbb{I}}\mathring{\mathbb{I}}\mathring{\mathbb{I}})} \\ \rho_{(\mathbb{II},\mathring{\mathbb{I}})} & \rho_{(\mathbb{II},\mathbb{D})} & \rho_{(\mathbb{II},\mathring{\mathbb{I}}\mathring{\mathbb{I}}\mathring{\mathbb{I}})} & \rho_{(\mathbb{II},\mathring{\mathbb{I}}\mathring{\mathbb{I}})} \\ \rho_{(\mathbb{II},\mathring{\mathbb{I}}\mathring{\mathbb{I}})} & \rho_{(\mathbb{II},\mathring{\mathbb{I}}\mathring{\mathbb{I}}\mathring{\mathbb{I}})} & \rho_{(\mathbb{II},\mathbb{D})} & \rho_{(\mathbb{II},\mathring{\mathbb{I}})} \\ \rho_{(\mathbb{II},\mathring{\mathbb{I}}\mathring{\mathbb{I}}\mathring{\mathbb{I}})} & \rho_{(\mathbb{II},\mathring{\mathbb{I}}\mathring{\mathbb{I}})} & \rho_{(\mathbb{II},\mathring{\mathbb{I}})} & \rho_{(\mathbb{II},\mathbb{D})} \end{pmatrix};$$

if $\Gamma_{ab} = \mathbb{III}$, then

$$\mathbf{\Omega}^{ab} = \mathbf{\Omega}(\mathbb{III}) = \begin{pmatrix} \rho_{(\mathbb{III},\mathbb{D})} & \rho_{(\mathbb{III},\mathring{\mathbb{I}})} & \rho_{(\mathbb{III},\mathring{\mathbb{I}}\mathring{\mathbb{I}})} & \rho_{(\mathbb{III},\mathring{\mathbb{I}}\mathring{\mathbb{I}}\mathring{\mathbb{I}})} \\ \rho_{(\mathbb{III},\mathring{\mathbb{I}})} & \rho_{(\mathbb{III},\mathbb{D})} & \rho_{(\mathbb{III},\mathring{\mathbb{I}}\mathring{\mathbb{I}}\mathring{\mathbb{I}})} & \rho_{(\mathbb{III},\mathring{\mathbb{I}}\mathring{\mathbb{I}})} \\ \rho_{(\mathbb{III},\mathring{\mathbb{I}}\mathring{\mathbb{I}})} & \rho_{(\mathbb{III},\mathring{\mathbb{I}}\mathring{\mathbb{I}}\mathring{\mathbb{I}})} & \rho_{(\mathbb{III},\mathbb{D})} & \rho_{(\mathbb{III},\mathring{\mathbb{I}})} \\ \rho_{(\mathbb{III},\mathring{\mathbb{I}}\mathring{\mathbb{I}}\mathring{\mathbb{I}})} & \rho_{(\mathbb{III},\mathring{\mathbb{I}}\mathring{\mathbb{I}})} & \rho_{(\mathbb{III},\mathring{\mathbb{I}})} & \rho_{(\mathbb{III},\mathbb{D})} \end{pmatrix}.$$

The integrative correlation matrix is then

$$R = \begin{pmatrix} \mathbf{\Omega}^{11} & \cdots & \mathbf{\Omega}^{19} \\ \vdots & \ddots & \vdots \\ \mathbf{\Omega}^{91} & \cdots & \mathbf{\Omega}^{99} \end{pmatrix}.$$

To apply the proposed MTLC model, all OTU observations are summarized as $Y$. $X$ is the single binary predictor denoting obesity status (lean vs. obese/overweight). Both $Y$ and $X$ have dimension $K \times J$ where $K = 54$ and $J = 36$. Some pedigrees only consist one individual instead a pair of twins, and OTUs are observed at one instead of two time points for some individuals, hence missing values exist in the matrix $Y$. Next, $Y$ is separated as $Y^{(0)}$ and $Y^{(+)}$ representing OTU prevalences and positive RAs. We assume each $y_{kj}^{(0)}$ follows Bernoulli distribution with mean $\mu_{kj}^{(0)}$ and $y_{kj}^{(+)}$ follows log normal distribution with mean $\mu_{kj}^{(+)}$. Then under MTLC model,

**Table 5. Estimated effects of obesity status to OTUs and p-value.**

|  | GEE$^{(0)}$ | GEE$^{(+)}$ | MTLC | 2P_ind | 1P_GEE | 1P_ind | 1P_RE |
|---|---|---|---|---|---|---|---|
| $\hat{\beta}$ | NA | NA | NA | NA | -0.041 | -0.024 | -0.028 |
| $\hat{\beta}^{(0)}$ | -0.511 | NA | -0.511 | -0.496 | NA | NA | NA |
| $\hat{\beta}^{(+)}$ | NA | -0.017 | -0.017 | 0.014 | NA | NA | NA |
| p-value | 0.017 | 0.518 | 0.034 | 0.093 | 0.215 | 0.450 | 0.475 |

$Y$ and $X$ have the following relationship:

$$\log\left(\frac{\mu_{kj}^{(0)}}{1-\mu_{kj}^{(0)}}\right) = \alpha^{(0)} + x_{kj}^{(0)}\beta^{(0)} \tag{8}$$

$$\mu_{kj}^{(+)} = \alpha^{(+)} + x_{kj}^{(+)}\beta^{(+)} \tag{9}$$

$\alpha^{(0)}$ and $\alpha^{(+)}$ are intercept parameters which are not our primary interest. Our goal is to estimate the effects of obesity status $\beta^{(0)}$ and $\beta^{(+)}$, and test $H_0$: $\beta^{(0)} = \beta^{(+)} = 0$. $\beta^{(0)}$ and $\beta^{(+)}$ are estimated separately under Eq 2, and $H_0$ is tested by the combined test statistic $W_{MTLC}$ following Eq 7.

We summarize the estimates of obesity effects for predicting OTUs and corresponding p-values for testing $H_0$ in Table 5. We compare the MTLC model with the other models listed in Table 1. Using our MTLC model, obesity has shown significant overall association with these OTUs. Specially, it has shown significant association with the prevalence of OTUs, but no significant association with the non-zero RAs. All other models do not detect the overall significance. The computation time is less than 30 seconds for the twin study dataset.

Correlation estimates are presented in Table 6. $\rho_{(\mathbb{D},\mathbb{I})}$ and $\rho_{(\mathbb{D},\mathbb{II})}$ are correlation between the two time points and correlation between the two twins. $\rho_{(\mathbb{I},\mathbb{D})}$, $\rho_{(\mathbb{II},\mathbb{D})}$ and $\rho_{(\mathbb{III},\mathbb{D})}$ are OTU

**Table 6. Estimated correlation coefficients between time points, twins and OTUs.**

| Models | | GEE | Pearson |
|---|---|---|---|
| GEE$^{(0)}$ | $\rho_{(\mathbb{D},\mathbb{I})}$ | 0.098 | 0.106 |
| | $\rho_{(\mathbb{D},\mathbb{II})}$ | 0.130 | 0.110 |
| | $\rho_{(\mathbb{I},\mathbb{D})}$ | 0.229 | NA |
| | $\rho_{(\mathbb{II},\mathbb{D})}$ | 0.217 | NA |
| | $\rho_{(\mathbb{III},\mathbb{D})}$ | 0.347 | NA |
| GEE$^{(+)}$ | $\rho_{(\mathbb{D},\mathbb{I})}$ | 0.696 | 0.751 |
| | $\rho_{(\mathbb{D},\mathbb{II})}$ | 0.550 | 0.561 |
| | $\rho_{(\mathbb{I},\mathbb{D})}$ | -0.018 | NA |
| | $\rho_{(\mathbb{II},\mathbb{D})}$ | -0.035 | NA |
| | $\rho_{(\mathbb{III},\mathbb{D})}$ | -0.175 | NA |
| 1P_GEE | $\rho_{(\mathbb{D},\mathbb{I})}$ | 0.661 | 0.657 |
| | $\rho_{(\mathbb{D},\mathbb{II})}$ | 0.495 | 0.498 |
| | $\rho_{(\mathbb{I},\mathbb{D})}$ | 0.051 | NA |
| | $\rho_{(\mathbb{II},\mathbb{D})}$ | 0.082 | NA |
| | $\rho_{(\mathbb{III},\mathbb{D})}$ | 0.015 | NA |

correlations, representing correlation from different family but within the same order *Clostridiales*, and correlation within the same family *Lachnospiraceae* or *Ruminococcaceae*.

When Pearson correlations are available ($\rho_{(\mathbb{D},\hat{\imath})}$ and $\rho_{(\mathbb{D},\hat{\text{iii}})}$), they are quite consistent with the correlation estimates under GEE models. However, Pearson correlation is not available for OTU correlations due to the complicated taxonomic structure, and only our proposed MTLC model can estimate these correlations.

## Discussion

In this paper, we develop and implement a novel approach to model the correlations of OTUs based on the biological taxonomic structure. The proposed MTLC model can incorporate the taxonomic structure with repeated measures from longitudinal data. It has accurate Type I error, unbiased estimation of model parameters and robust power performance under a variety of situations. Compared to existing methods, our method is more powerful and can provide unbiased estimation of the correlation coefficients between multiple OTUs and repeated measures.

The MTLC model allows for sufficient flexibility of the correlation matrix construction. It not only allows different correlation matrices for the logistic regression part and linear regression part, but also put no constraint on the range of each correlation coefficient, i.e., any positive or negative value from -1 to 1. In contrast, the random effect in mixed effect model naturally leads to a positive correlation, because the same random effect adds to a few correlated samples. When the true correlations are negative, the mixed effects model (e.g., Chen et al. [13]) is simply reduced to ordinary linear and logistic regression model with independence assumption, which results in incorrect Type I errors as we have shown in section "Inferences for predictor's main effects". In summary, the MTLC model provides a reliable analytical framework for longitudinal microbiome data analysis.

Our methodology for constructing correlation matrix of taxonomic structure imposes no constraints to the number of OTUs, which is denoted by $N$. Based on the computation time shown in our simulation and application study, we find the MTLC model runs fast overall. However, when $N$ is large, (e.g., $N > 1000$), the correlation matrix has a high dimension, and it may cause computational issues and become time consuming to implement the MTLC model. In such case, we suggest a dimension reduction by selecting a subgroup of OTUs. For example, if OTUs are from the same phylum but different classes. Our MTLC model can be implemented on each class separately or focus on the classes of interest, instead on the whole phylum.

We have shown that the correlation estimation is consistent under MTLC model, but the estimation accuracy is not clear. Yan [36] proposed standard error estimations of the correlation coefficients under GEE approach. When corresponding Pearson correlations are also available, we have found the standard error under GEE approach may depart from the standard error of Pearson correlations. Because the underlying distribution of the correlation estimates is unknown, it lacks theoretical justifications of the standard error estimates. Further studies are required for estimating the accurate standard errors of correlation coefficients under our MTLC model.

The MTLC model assumes $-\log_{10}$ transformed positive RAs following normal distribution. Clearly this is not the only approach to modelling the RA data, and there is no universal answer for choosing the "best" approach. Liu et al. [38] gave an overview for modelling zero-inflated non-negative continous data in general and proposed a few alternative distributions for the positive part of RAs. For example, zero-inflated beta distribution is another commonly

used approach [13, 39], because beta distribution has range from 0 to 1 exactly matching the range of RAs.

When $\beta^{(0)}$ and $\beta^{(+)}$ have opposite signs, the predictor's effects are described as "dissonant". Under this scenario, the two-part models showing more powerful results in the simulation studies coincides with existing literature [9, 40]. In microbiome context, an example of this scenario is that, an antibiotic treatment may be effective in reducing the risk of carrying some specific bacteria, but may result in the growth of these bacteria once they survive due to antibiotic resistance [41, 42].

For the proposed method, the dimension of predictors' design matrix $X_k$, $p$, is assumed to be less than the number of clusters $K$. For high dimensional predictor space, e,g., gene expressions in genome-wide association study, it is possible to encounter the situation of $p \geq K$. In such cases regression models cannot be directly applied, and dimension reduction techniques need to be used. Traditional approaches such as principal component analysis and penalized regression including ridge regression and LASSO, as well as some machine learning based feature selection methods can be considered to be incorporated into the proposed method to deal with high dimensional predictors. We are planning to extend the proposed method to deal with such high dimensional predictors situation.

We have treated repeated longitudinal measures as a few discrete time points in our MTLC model. When there are more time points for each sample and the exact observation time for each sample is continuous, it is a natural extension of our current work to consider time as a continuous variable and OTU observations as a function of time. Further investigation of functional data analysis techniques can be explored and integrated with the OTU correlation structure developed in this paper.

## Supporting information

**S1 Data. Data for the real microbiome sequencing study in Application section.**
(XLS)

**S1 Appendix. Additional simulation results.**
(PDF)

## Acknowledgments

The authors would like to thank Dr. Lillian L. Siu, Dr. Bryan Coburn, Dr. Pierre Schneeberger, Dr. Osvaldo Espin-Garcia and Dr. Jeffrey Rosenthal for helpful discussions and suggestions at different stages of our study.

## Author Contributions

**Conceptualization:** Wei Xu.

**Formal analysis:** Bo Chen.

**Funding acquisition:** Wei Xu.

**Investigation:** Bo Chen, Wei Xu.

**Methodology:** Bo Chen, Wei Xu.

**Supervision:** Wei Xu.

**Writing – original draft:** Bo Chen.

**Writing – review & editing:** Bo Chen, Wei Xu.

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
