## [Decision Letter · Decision Letter 0]

2 May 2020

Dear Dr. Xu,

Thank you very much for submitting your manuscript "Generalized Estimating Equation Modeling on Correlated Microbiome Sequencing Data with Longitudinal Measures" for consideration at PLOS Computational Biology.

As with all papers reviewed by the journal, your manuscript was reviewed by members of the editorial board and by several independent reviewers. In light of the reviews (below this email), we would like to invite the resubmission of a significantly-revised version that takes into account the reviewers' comments.

We cannot make any decision about publication until we have seen the revised manuscript and your response to the reviewers' comments. Your revised manuscript is also likely to be sent to reviewers for further evaluation.

Sincerely,

Benjamin Althouse

Associate Editor

PLOS Computational Biology

Jason Papin

Editor-in-Chief

PLOS Computational Biology

Reviewer's Responses to Questions

**Comments to the Authors:**

Reviewer #1: In this article, the authors present an innovative method to analyze microbiome data for disease prediction and correlation analysis. It is very important to build a statistical and computational model which fully accounts for the correlation relationships among the OTUs.

The theoretical investigations are rigorous and the numeric studies provide very comprehensive assessment of the empirical performance of the proposed methods. The paper is well organized and all the theoretical and numerical results are presented in a very concise and rigorous way.

The authors proposed to use a two-part Microbiome Taxonomic Longitudinal Correlation (MTLC) model for multivariate zero-inflated OTU outcomes based on the GEE framework. Longitudinal and other types of repeated OTU measures are integrated in the MTLC model. Variance estimators of the proposed regression estimates are fully developed. Compared

with the existing methods including traditional GEE and mixed models, the MTLC method is shown to be more powerful and more accurate in numerical studies. Authors have also investigated the performance of the method on a real microbiome data.

Compared to existing predictive methods, the newly proposed method is advantageous for microbiome data analysis because it models the correlation structure among the repeated measurements and among the correlated OTUs. The authors have provided innovative and significant contributions in this paper. I strongly recommend the publication of this paper.

A minor comment is regarding the dimension of the predictor. It would be helpful to the readers if the authors can discuss how the model can be extended to accommodate higher dimensional predictors which often arises in practical applications.

Reviewer #2: In this paper, the authors proposed a Microbiome Taxonomic Longitudinal Correlation (MTLC) model to test the association between operational taxonomic units (OTUs) and the predictors of interest. The model consists of two parts: a logistic regression of the OTU prevalence and a linear regression of OTU relative abundance (RA). An omnibus test of these two regressions was developed to assess overall statistical significance. The model parameters and their variances are estimated by Generalized Estimating Equation (GEE) approach to account for the correlation between the OTUs and repeated measures. The simulation study shows that the proposed method can control type I error at the nominal level 0.05 and the power is the most robust against different configurations of the effect sizes. The advantage of the proposed method is further evidenced by a real data analysis in which the association between obesity and OTUs was detected by the proposed method but not by other comparative methods.

The idea of applying GEE to model longitudinal/correlated data is not new and but a two-part GEE with estimated correlation matrix seems to be novel in microbiome studies. However, some major concerns about the proposed statistical model, the simulation settings and the power comparison results need to be addressed.

General:

1. The authors mention in the Abstract and Introduction that the proposed method is able estimate the correlations between OTUs. However, the benefits of obtaining those correlation coefficients are not clearly stated. What additional information can an accurate estimated correlation bring to us?

2. The presented bibliography is rich. However, previous applications of GEE in microbiome data were not mentioned, e.g.

Kelly, B.J., Imai, I., Bittinger, K. et al. Microbiome 4, 7 (2016). https://doi.org/10.1186/s40168-016-0151-8

Seekatz, A.M., Rao, K., Santhosh, K. et al. Genome Med 8, 47 (2016). https://doi.org/10.1186/s13073-016-0298-8.

Statistical Model:

1. The statistical model is confusing. The authors seem to assume that, within each independent block, y and x are linked by a GLM (1). The y0, i.e. the dichotomized y, and y+, i.e. the truncated y>0, relate to x by other GLMs (5) and (6), respectively. If so, the distribution of y can be and should be stated explicitly. If not, what is the relationship between y and (y0, y+)? This also concerns the data generating distribution in the simulation. See the comments regarding simulations.

2. The authors claim that the test statistics W0 and W+ are independent (P.9 line 211). I am not convinced this is true for arbitrary y as it seems to be presented in the paper. In fact, given a vector y, the number of non-zero elements, i.e. E(y0), determines the length of y+ which seems to contradict the independence claim. The authors should clarify under what kind of distribution of y this independence property holds.

3. An omnibus test W_MTLC (7) is proposed to combine W0 and W+ by summation. Other approaches can be used to combine the p-values of these two tests, e.g. the minimum P-value (minP) approach and Cauchy combination test (CCT) Yaowu Liu & Jun Xie (2020) JASA, 115:529, 393-402, DOI: 10.1080/01621459.2018.1554485. Notice that the power of W0 and W+ can be drastically different in the simulations, I would expect the power of either minP or CCT to be higher than the summation of statistics as proposed.

Simulation:

1. The data generating distribution/process (P.9 lines 239-246 and P.9 lines 252-256) should be clearly written in statistical symbols/equations to avoid confusion. Most importantly, how are Y0, Y+ and Y related to the predictor X?

2. It seems that this simulation setting is fundamentally different from the simulations in literature, e.g. [11], [16], in which the distribution of Y is clearly defined, while in this paper, Y is constructed by Y0 and Y+. It would be very helpful if authors can

2.1. explain why they choose such way to simulate Y0 and Y+ and Y.

2.2. explain how to interpret the effect sizes beta0 and beta+. Are we particularly interested in detecting a pair of (beta0, beta+) in different directions (as shown in Table 2, this is where the proposed method has the largest power gain) in real data?

2.3. simulate Y from a simpler model, e.g. a linear mixed model, as a sensitivity analysis to see if the proposed method still works fine.

3. Table 2. Estimated beta, Type I error and power:

3.1. The authors state that when beta0 and beta+ are in the same direction, the two-parts model is still more powerful in “general” (P.13 lines 307-308). This is simply not true. In Table 2, there are only two rows (row 6 and 10) where effects are in the same direction. In both cases, the 1P_GEE and 1P_RE both have higher power than MTLC.

3.2. The authors should make it clear that the estimated beta, beta0, beta+ are from one simulation or the average from 1,000 replication? A more sensible way is to report the empirical distribution of the estimated beta, e.g. standard deviation, 2.5 and 97.5 percentile of the empirical distribution.

Minor (typos, etc.):

1. Author Summary and P.1 line 2: “…fast-growing…”.

2. P.2 lines 5-6: “…shotgun metagenomics sequencing…”.

3. P.5 line 106: “…other entries…”.

4. P.8 line 185, P11 line 282: “Next, we…”.

5. P.9 lines 196-197: mu0 (5) and mu+ (6) are not defined.

6. P.10 line 231: “…may be greater than…”.

7. P.12 Table 2: (beta_B, beta_N) should be (beta0, beta+).

8. P.14 line 363: “…one-part models…”.

9. P.15 line 375: “… at level genus.”

10. P.18 line 441: “…distribution…is unknown…”

**Have all data underlying the figures and results presented in the manuscript been provided?**

Reviewer #1: Yes

Reviewer #2: None

PLOS authors have the option to publish the peer review history of their article (what does this mean?). If published, this will include your full peer review and any attached files.

Reviewer #1: No

Reviewer #2: No
---

## [Decision Letter · Decision Letter 1]

30 Jun 2020

Dear Dr. Xu,

We are pleased to inform you that your manuscript 'Generalized Estimating Equation Modeling on Correlated Microbiome Sequencing Data with Longitudinal Measures' has been provisionally accepted for publication in PLOS Computational Biology.

Best regards,

Benjamin Althouse

Associate Editor

PLOS Computational Biology

Jason Papin

Editor-in-Chief

PLOS Computational Biology

Reviewer's Responses to Questions

**Comments to the Authors:**

Reviewer #1: The authors have addressed the questions raised in my reports. I recommend the acceptance of the paper.

Reviewer #2: The authors have addressed all the concerns. I don't have further comments.

**Have all data underlying the figures and results presented in the manuscript been provided?**

Reviewer #1: Yes

Reviewer #2: None

PLOS authors have the option to publish the peer review history of their article (what does this mean?). If published, this will include your full peer review and any attached files.

Reviewer #1: No

Reviewer #2: No

---

## [Editor Report · Acceptance letter]

19 Aug 2020

PCOMPBIOL-D-20-00510R1 

Generalized Estimating Equation Modeling on Correlated Microbiome Sequencing Data with Longitudinal Measures

Dear Dr Xu,

I am pleased to inform you that your manuscript has been formally accepted for publication in PLOS Computational Biology. Your manuscript is now with our production department and you will be notified of the publication date in due course.

With kind regards,

Sarah Hammond
